# Effect of pH-Shifting Process on the Cathepsin Activity, Muddy Off-Odor Compounds’ Content and Gelling Properties of Isolated Protein from Silver Carp

**DOI:** 10.3390/foods12050939

**Published:** 2023-02-22

**Authors:** Weidan Guo, Miao Zhan, Hui Liu, Xiangjin Fu, Wei Wu

**Affiliations:** 1College of Food Science and Engineering, Central South University of Forestry and Technology, Changsha 410004, China; 2Hunan Provincial Key Laboratory of Processed Food for Special Medical Purpose, Changsha 410004, China; 3Hunan Provincial Engineering Technology Research Center of Seasonings Green Manufacturing, Changsha 410004, China

**Keywords:** silver carp (*Hypophthalmichthys molitrix*), pH-shifting process, muddy off-odor, gelling properties, cathepsin

## Abstract

Silver carp (*Hypophthalmichthys molitrix*) is a potential source for making surimi products. However, it has the disadvantages of bony structures, high level of cathepsines and muddy off-odor which is mainly caused by geosmin (GEO) and 2-methylisoborneol (MIB). These disadvantages make the conventional water washing process of surimi inefficient (low protein recovery rate, and high residual muddy off-odor). Thus, the effect of the pH-shifting process (acid-isolating process and alkali-isolating process) on the cathepsins activity, GEO content, MIB content, and gelling properties of the isolated proteins (*IPs*) was investigated, comparing it with surimi obtained through the conventional cold water washing process (WM). The alkali-isolating process greatly boosted the protein recovery rate from 28.8% to 40.9% (*p* < 0.05). In addition, it removed 84% GEO and 90% MIB. The acid-isolating process removed about 77% GEO and 83% MIB. The acid-isolated protein (AC) displayed the lowest elastic modulus (G′), the highest TCA-peptide content (90.89 ± 4.65 mg/g) and the highest cathepsin L activity (65.43 ± 4.91 U/g). The AC modori (60 °C for 30 min) gel also demonstrated the lowest breaking force (226.2 ± 19.5 g) and breaking deformation (8.3 ± 0.4 mm), indicating that proteolysis caused by the cathepsin deteriorated the gel quality of AC. The setting (40 °C for 30 min) considerably increased the breaking force (386.4 ± 15.7 g) and breaking deformation (11.6 ± 0.2 mm) of the gel made from the alkali-isolated protein (AK) (*p* < 0.05). In AC and AK gel, a clearly visible cross-linking protein band with a molecular weight greater than MHC was seen, demonstrating the presence of endogenous trans-glutaminase (TGase) activity, that improved the gel quality of AK. In conclusion, the alkali-isolating process was an effective alternative method for making water-washed surimi from silver carp.

## 1. Introduction

The huge annual yield and high nutritional value (rich in protein and polyunsaturated fatty acids, PUFAs) of silver carp (*Hypophthalmichthys molitrix*) are the main reasons for its global importance [1]. With the increase in the global population and the lack of protein resources, researchers have paid attention to the utilization of fish protein resources, such as the carps in fresh water in the United States [1]. Yet, due to its bony form and muddy off-odor, the commercial value of fresh silver carp is quite restricted [2]. The small bones also make the typical mechanical approach of meat recovery from carp inefficient [3,4].

Fortunately, the pH-shifting process developed by Hultin and Kelleher [5] is convenient for recovering protein from bony fish. This new method involves dissolving fish muscle protein at either low pH (2–3) or extremely high pH (10.5–12), removing small bones using high force centrifugation or filtering, and then adjusting the pH of the liquid to neutrality (pH 5–7) in order to recover the isolated protein (IP). This method has been carried out in catfish (*Silurus asotus*) [6,7], atlantic menhaden *(Brevoortia tyrannus*) [8], giant squid (*Architeuthis*) [9], tilapia (*Oreochromis mossambicus*) [10,11,12], Cape hake (*Merluccius capensis*) [13], yellow stripe trevally (*Selaroides leptolepis*) [14], jumbo squid (*Dosidicus gigas*) [15], kilka (*Clupeonella cultriventris*) [16,17], etc.

Additionally, the functional characteristics of silver carp protein recovered via the pH-shifting technique were revealed in the literature. This new process successfully eliminated contaminants such as bones, scales, skin, and fins from whole gutted carp [2,3], and the IPs could produce gels that were as strong as Alaska pollock surimi or stronger [18]. The principal gelling protein in muscle, myosin, was evaluated for its conformation and denaturation using differential scanning calorimetry (DSC), and the data showed that greater transition temperatures and myosin’s enthalpy resulted in improved gelling qualities [2]. However, no literature has focused on the effect of the pH-shifting process on several other components such as cathepsins and trans-glutaminase (TGase) which also determine the gel properties of surimi.

The high activity of cathepsins is a major disadvantage of silver carp muscle for gel preparation [19,20], because cathepsins degrade myosin during the gelling temperature of 40–65 °C, resulting in a decrease in gel strength (called “modori”). In contrast, TGase could improve the gel strength by catalyzing formed cross-links (bonds of ε-(y-glutamyl) lysine) between proteins at 4–40 °C. Microbial TGase was found to really improve the mechanic properties of silver carp mince and surimi [4,21]. The pH-shifting process recovers a lot of sarcoplasmic proteins, including the cathepsines and TGase, but they might denature at different degrees in the extreme pH conditions of the pH-shifting process, surely affecting the gelling properties of IPs in different ways.

A muddy off-odor is another disadvantage of silver carp muscle being used as food. A muddy off-odor in silver carp is caused mainly by geosmin (GEO) and 2-methylisoborneol (MIB), with a threshold of about 0.7 μg/kg [22]. Due to their lipophilic and hemi-volatile nature, they are exceedingly difficult to be removed via traditional processes, such as washing, drying, curing and fermentation [22]. The muddy off-odor in catfish and tilapia was reportedly removed extremely effectively [6,7] using the pH-shifting technique [23]. There is no information available regarding how this treatment affects the silver carp’s muddy off-odor.

Thus, it is very interesting to look at the impact of pH-shifting on the cathepsin activity, TGase activity, GEO content and MIB content in silver carp. Additionally, the gelation properties of IPs were also determined in this study.

## 2. Materials and Methods

### 2.1. Materials

Silver carp fillets were bought from a local supermarket. Fresh silver carps (1 kg average) were slaughtered, headed, gutted, filleted and washed (4 °C) manually.

All chemical reagents used were analytical grade and were purchased from Sinopharm Chemical Reagent Co., Ltd. (Shanghai, China) or Sigma Aldrich Co. (St. Louis, MO, USA).

### 2.2. Methods

#### 2.2.1. Mince Collection

Skinless, boneless mince for the conventional water washing process was obtained by passing the fillet through a mini belt deboner (Hengchang Machinery Manufacture Co., Ltd., Linyi, China). The aperture of the machine was 5 mm (diameter).

The fillets for the pH-shifting process were removed from the back bone and skin manually (regardless of the small bones and connective tissue), and then cut into slices.

#### 2.2.2. Preparation of Conventional Water-Washed Surimi

The water-washed surimi (WM) was prepared according to Luo et al. (2008) [4]. The mince was mixed three times (v/w) with cold water (4 °C), stirred for 3 min and left to stand for 15 min (4 °C). Then, we hand-extruded and filtered the washed meat with three layers of coarse cotton. Washing and filtration processes were repeated three times. The water-washed surimi was wrapped in polyethylene bags and stored on ice in a cold closet until its use on the same day.

#### 2.2.3. The pH-Shifting Process

The procedure was performed according to Hultin [5] with slight modification. Fish slices were homogenized (DS-1 high speed homogenizer, Sample’s Model Factory, Shanghai, China) 9 times (*v*/*w*) with cold water at 10,000 r/min for 30 s. Then, the meat slurry was stirred and adjusted the pH value using HCl or NaOH solution (1 mol/L). The pH values were 11.8 and 2.3 for alkali- and acid-isolating processes, respectively. After adjusting the pH to the desired (expected) value, it was continuously stirred for 30 min, followed by centrifuging (4 °C, 10,000× *g* for 10 min), and discarded the sediment (small bones, connective tissue and scales), adjusted the pH value of the supernatant to 5.5 using HCl or NaOH solution (1 mol/L) and then centrifuged (4 °C, 10,000× *g* for 30 min). The supernatant was discarded, and the isolated proteins (IPs) in the sediment, including the acid-isolated protein (AC) or alkali-isolated protein (AK), were collected and packed in polyethylene bags, stored in an ice-cold closet and used within four hours.

#### 2.2.4. Protein Recovery Rate

After the fish fillets and IPs were recovered, the weight was recorded and the weight ratio was calculated, the ratio was the recovery rate (at the same moisture content). The recovery of the protein was calculated as follows:$$\mathrm{Protein}\;\mathrm{recovery}\;(\%)=\frac{\mathrm{Weight}\;\mathrm{of}\;\mathrm{recovered}\;\mathrm{protein}}{\mathrm{Weight}\;\mathrm{of}\;\mathrm{initial}\;\mathrm{fillet}}\times 100$$

#### 2.2.5. Cathepsin Activity

Samples were extracted with 25 mmol/L sodium acetate (*v*/*w*) buffer solution which contains 5 mmol/L cysteine and 0.3 mmol/L phenylmethylsulfonyl fluoride (PMSF), and the pH of the extraction was adjusted to 5.0, then centrifuged at 12,000× *g* for 2 min and repeated 4 times. The supernatant was collected as the enzyme extraction.

Enzyme activity was determined according to Liu et al. [20]. Briefly, Z-Phe-Arg-MCA and Z-Arg-Arg-MCA were used as substrates for cathepsin L and cathepsin B, respectively. The enzyme and substrate were blended, and then incubated at 40 °C for 10 min. The fluorescent intensity of aminomethylcoumarin (AMC) released via hydrolysis was determined in a fluorescence spectrophotometer (LS55, Perkin Elmer, Waltham, MA, USA) with 380 nm of excited wavelength and 460 nm of emitted wavelength. The amount of activity releasing 1 nmol of AMC per minute was recorded as one unit of enzyme activity.

#### 2.2.6. Determination of GEO and MIB

The GEO and MIB were extracted based on the method of Phetsang [6] and Fu [24]. Microwave-mediated distillation and solid-phase microextraction (SPME) were used for extraction, followed by gas chromatography mass spectrometry (GC/MS) for analysis, using the PEG-20M capillary column (30 m × 0.25 mm × 0.25 μm). The helium carrier gas (99.99% purity) flow rate was 1.0 mL/min. The initial temperature of the GC oven was 60 °C, then it increased to 120 °C at 10 °C/min rate, then it increased to 230 °C at 20 °C/min rate and it was held for 1 min. The injector temperature was 200 °C, and the ionization energy (EI) was 70 eV; the EI ionization source was 230 °C. The detector was set in the electron impact (EI) ionization mode at 70 eV in full scan mode with a mass/charge ratio (*m*/*z*) range from 50 to 500. The characteristic ion mass/charge ratio (*m*/*z*) of MIB and GEO was 95 and 112, respectively.

Gas chromatograph grade MIB (Sigma Aldrich Co., Ltd., St. Louis, MO, USA) and GEO were used as the exterior standards.

#### 2.2.7. Rheology Studies

The function of temperature in the gel forming procedure was determined using an AR-1000 rheometer (TA Instruments, New Castle, DE, USA). The WM and IPs were diluted with 0.60 mol/L KCl and 0.02 mol/L Tris-HCl buffer (pH 7.0) at 4 °C, until the final protein concentration was 35 mg/mL. The solution was then homogenized at 10,000 r/min for 30 s, before the air bubbles were removed via vacuuming. The conditions of the rheometer were as follows: gap 1 mm, steel cone 2°, shear strain 0.02, shear frequency 0.1 HZ and the temperature increased from 20 °C to 85 °C at 1.5 °C/min.

#### 2.2.8. Heat-Induced Gel

The WM and IPs were chopped by a food processor (La Minerva 22A/22L, Bologna, Italy) (10 °C) for 6 min, and then sucrose (2 g/100 g), sorbitol (2 g/100 g), NaCl (3 g/100 g) and polyphosphate sodium (0.3 g/100 g) were added, with the addition of crushed ice and the moisture content being adjusted to 80 g/100 g. The paste was put into a polyvinylidine chloride casing with a diameter of 2.5 cm and was sealed tightly at both ends. The paste was incubated directly at 85 °C for 30 min (kamaboko gel), or at 40 °C (setting gel) or 60 °C (modori gel) for 30 min, and was then heated at 85 °C for 30 min in a water bath [8]. After all of the gels were heated, they were immediately cooled in ice water for 30 min, and then stored at 4 °C for analysis the next day.

#### 2.2.9. Determination of the Gel Strength: Hardness and Elasticity

Gel hardness and elasticity were determined by a puncture test using a TA-XT2 texture analyzer (Stable Micro Systems, Godalming, UK) which was equipped with a 5 mm cylinder plunger (P/0.5). Gels were evaluated and equilibrated at ambient temperature (28–30 °C) for 2 h. Five cylinder-shaped samples (d = 2 cm) which had a length of 2.5 cm were measured. When the gel was compressed with a speed of 60 mm/min and operating depth of 2 cm, the load value (g) at the broken point indicated the hardness, while the breaking deformation (mm) represented the elasticity [25].

#### 2.2.10. TCA-Soluble Peptide Measurement

TCA-soluble peptide was determined according to the Lowry method and the content was expressed as a mg tyrosine/g sample [25].

#### 2.2.11. Determination of Disulfide Bond Content

The disulfide bond was assayed using 5,5′-dithiobis-(2-nitrobenzoic acid) (DTNB) according to the method of Benjakul [26]. The disulfide bond was calculated using the extinction coefficient of 13,900/mol/cm.

#### 2.2.12. SDS-Polyacrylamide Gel Electrophoresis

Gel electrophoresis was carried out according to Laemmli [27] using 4% stacking gel and 10% acrylamide separating gel. The sample buffer was triseglycine (pH 8.8), which contained 1 g/100 mL SDS and 1 mL/100 mL L-mercaptoethanol. The sample concentration was 1 mg/mL, and the loading volume was 15 μL. Electrophoresis (Bio-RAD Mini Protein system, Bio-Rad, Hercules, CA, USA) was carried out at a constant voltage of 120 V. The film was fixed with 15% trichloroacetic acid for 30 min, dyed with 0.125% (*w*/*w*) R-250 for 12 h and then decolorized with 40% ethanol (*v*/*v*) + 10% acetic acid (*v*/*v*).

#### 2.2.13. Statistical Analysis

All chemical analyses were carried out in triplicate. In physical analyses, e.g., structure property, at least 5 measurements for each treatment were conducted. Data were subjected to analysis of variance analysis (ANOVA) using an SPSS package (SPSS 17.0 for Windows, SPSS Inc., Chicago, IL, USA).

## 3. Results and Discussion

### 3.1. Protein Recovery, Activity of Cathepsin, Content of GEO and MIB

As shown in Table 1, the protein recovery rates of the water washing process and alkali- and acid-isolating processes were 28.8%, 38.5% and 40.9%, respectively. The pH-shifting method extended the recovery rate considerably (*p* < 0.05) and the consequences were similar to the preceding studies of other researchers [1,2] because the water soluble sarcoplasmic proteins were recovered via a new method but went away with the water via the conventional water wash process. Moreover, numerous amounts of meat sticking to the small bones were discarded in the mechanical deboning of the conventional process but were recovered in the new process, which also increased the recovery rate.

As shown in Table 1, the cathepsins were effectively removed (*p* < 0.05). The muscle contained more cathepsin B than cathepsin L, while WM and IPs retained more cathepsin L than cathepsin B, indicating that cathepsin L bound more tightly with the muscle protein than with cathepsin B. The AC contained much higher activity of cathepsin L which might be due to the high activation of cathepsin L during acid treatment. Heidtmann et al. [28] showed that cathepsin L activity could be enhanced by acid treatment.

The muddy off-odor compounds in the silver carp muscle were effectively removed during the pH-shifting process by about 77% for GEO and 83% for MIB in the acid-isolating process, and 84% for GEO and 90% for MIB in the alkali-isolating process, respectively, while water washing removed about 39% of GEO and 50% of MIB (Table 1). This was similar with the results of Kleinholz Christina et al. [23], who found that the change process of the pH value could significantly reduce the odor of catfish (*p* < 0.001); MIB and GEO were reduced from 1.396 and 1.992 μg/kg to 0.104 μg/kg MIB and 0.258 μg/kg GEO, respectively, using phosphoric acid and to 0.0987 μg/kg MIB and 0.426 μg/kg GEO using NaOH. The removal of GEO and MIB was the result of the lipid reduction in the pH-shifting process, due to the lipophilic nature of GEO and MIB [7].

### 3.2. Rheology Properties

The elastic modulus (G′) of WM and IPs was shown in Figure 1. The G′ of WM included two peaks, at 47 °C and 76 °C, respectively, and a minimal peak at 57 °C, which was similar to the report of Abdollahi et al. [17]. The first G′ peak in the 40–50 °C range corresponded to the denature of the myosin head, with conformational changes and the exposure of reactive groups. This ensured a gradual sol–gel transition to form an ordered initial protein grid. Subsequently, the properties of light meromyosin changed and caused a decrease In G′, resulting in increased fluidity [29]. In addition, it has been increasingly accepted that the first increase in G′ involves the TGase-mediated covalent [26]. The second increase in G′ started at 57 °C, the peak was 76 °C and it corresponded to the myosin rod denature (at about 63 °C) and actin denature (at about 68 °C) [3]. In this step, the final kamaboko network was formed. In addition to the disulfide bonding, the formation of intermolecular hydrophobic interactions also occurs during heating [30,31].

The initial G′ of AC and AK was considerably lower than WM, indicating less molecule interaction in AC and AK. The G′ of AC and AK increased at about 42 °C, and the peak at 47 °C disappeared, indicating the denaturation or aggregation of the myosin head. According to Kristinsson and Hultin [32], the myosin partly unfolded in the extreme pH of the pH-shifting process, resulting in a “molten globular” state; when they then adjusted the pH to 5.5, the globulin rod refolded to the native structure, whereas the globulin head partially misfolded. Furthermore, it had to be noted that a typical ‘modori’ was observed for AC (the G′ of AC declined above 65 °C). The proteolytic degradation due to cathepsin enzymes was supposed to be the mechanism responsible for ‘modori’ [19,20], which is consistent with the result that AC retained the highest cathepsin L activity (Table 1).

### 3.3. The Content of TCA-Soluble Peptide

The TCA-soluble peptide is shown in Table 2. TCA-soluble peptides indicate the degree of hydrolysis of the muscle protein: the higher the content, the greater the degree of hydrolysis [25]. The TCA-soluble peptide content in WM was less than that of AC and AK (*p* < 0.05). In comparison to modori gels, the TCA-soluble peptide content of kamaboko gel was lower (*p* < 0.05). The highest TCA-soluble peptide content (90.89 mg/g) of the modori gel of AC indicates that serious hydrolysis of the proteins occurred. These results were consistent with the results of cathepsins (Table 1).

### 3.4. The Content of Disulfide

Numerous disulfide bonds were formed during gelling; however, there was no significant difference (*p* > 0.05) within the disulfide content of various gels. It is acknowledged that disulfide (S-S) bonds are crucial to the way that heat causes the protein to gel. SH oxidation into S-S bonds and/or the SH-induced S-S interchange could cause the covalent cross-linking of protein molecules. The disulfide bond is considered to be the main covalent bond formed in protein gel during high temperature heating (cooking at >40 °C). According to the report of Hossain et al. [33], the formation of disulfide bond polymerization in the WM gel was almost constant, and it was proposed that disulfide bond polymerization occurred during cooking at 80 °C.

### 3.5. Protein Pattern of WM, IPs and Gels

A decrease in the myosin heavy chain (MHC, 200 kDa) band intensity was determined in all modori gels compared with their corresponding kamaboko and setting gel (Figure 2). The lowest MHC band intensity was found in AC’s modori gel. The reduction in the MHC of AC gels was synchronal with an increase in TCA-soluble peptide content (Table 2), and AC retained more cathepsins (Table 1).

The gels of AC and AK showed obvious cross-linked protein bands with higher molecular weight than MHC (Figure 2). This might be due to the endogenous TGase, which can induce protein cross-linking, because the cross-linking of water-washed surimi gel proteins mainly includes the disulfide bond and TGase-mediated bond of ε-(y-glutamyl) lysine, and the disulfide content of gels was almost the same (Table 2). Because sarcoplasmic proteins are retained throughout the pH-shifting process, IPs might retain more endogenous TGase than conventionally washed surimi, and it was indicated that the extreme pH of the pH-shifting method did not totally inactivate the TGase. In addition, the conformational changes in the myofibrillar protein during the acid and alkaline processes could expose more functional groups, which were used for TGase-induced cross-linking [32].

### 3.6. Gel Strength

The breaking force and deformation force of kamaboko gel prepared from WM were higher than those from AC and AK, and the breaking force of the AC gel was the lowest (*p* < 0.05) (Table 3). These results were consistent with Paker’s report [3] about silver carp, finding that IP from the alkali-isolating process (solubilizing pH 11.5) showed significantly (*p* < 0.05) higher shear stress than IP from the acid-isolating process (solubilizing pH 2.5).

Although no peak was observed at the setting temperature (about 40 °C) in the G′ curve (Figure 1) for AK, setting improved the breaking force and deformation of the AK gel considerably (*p* < 0.05), as well as the WM gel, indicating that WM and AK contain notable TGase activity, in accordance with the result of SDS-PAGE (Figure 2). It has long been assumed that the activity of endogenous Tgase plays a significant role in setting. Tsukamasa et al. [34] reported that the index of breaking strength and the index of the (γ-glutamyl) lysine cross-link concentration had a high correlation (r = 0.987). The breaking force and breaking deformation of AC gels were dramatically reduced by Modori (*p* < 0.05), which might be explained by the fact that AC retained more cathepsins (Table 1). The proteolytic degradation of myofibrillar proteins, particularly myosin, resulted in a decrease in molecular mass and a loss in structural domains, which are essential for molecular interaction [35], such as forming gel structure. The proteolytic degradation of myofibrillar has an adverse effect on the quality of water-washed surimi and significantly reduces the gel strength. Taking into account the result of rheology (Figure 1) and cathepsin activity (Table 1), it was certain that the cathepsins retained in AC were key components in causing the gel deterioration.

## 4. Conclusions

The pH-shifting process increased the protein recovery rate and reduced the muddy off-odor compounds in silver carp effectively.

The AC retained the highest cathepsin L activity. Furthermore, the AC showed a higher content of the TCA-soluble peptide and a lower intensity of the MHC band, and the modori gel of AC demonstrated the lowest breaking deformation and breaking force which also proved the presence of cathepsins in AC.

Setting improved the breaking force and breaking deformation of AK gel considerably (*p* < 0.05), and an apparent cross-linking protein band with molecular mass on top of MHC was determined in AC and AK gels, indicating that the Tgase might be partly retained in the pH-shifting process. In conclusion, the alkali-isolating process was an effective alternative method for making surimi from silver carp.

## Figures and Tables

**Figure 1 foods-12-00939-f001:**
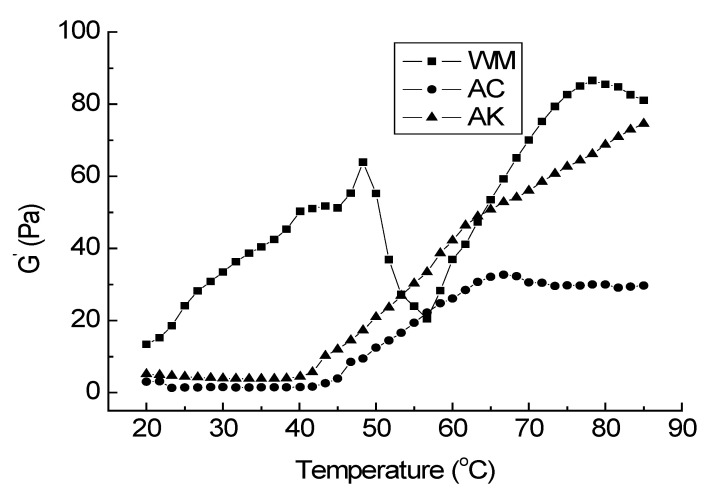
Rheology of surimi and isolated proteins. WM: water-washed surimi, AC: acid-isolated protein, AK: alkali-isolated protein.

**Figure 2 foods-12-00939-f002:**
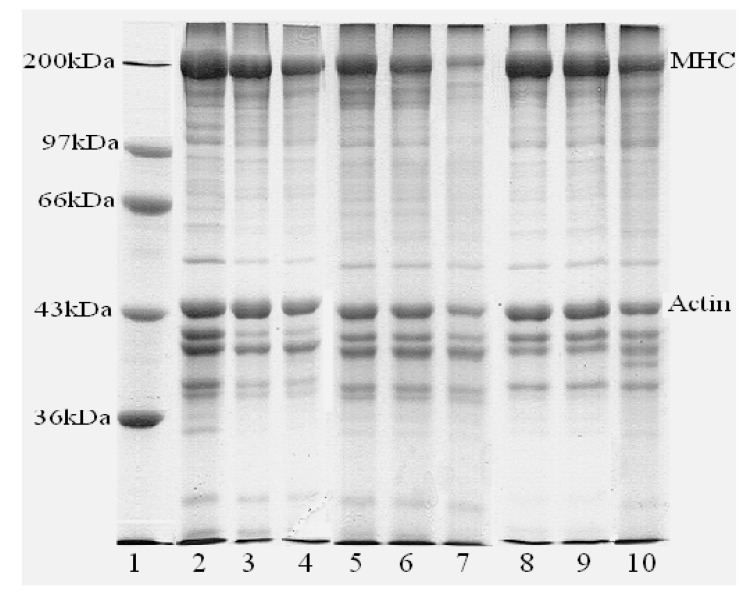
Protein patterns of gels prepared under different conditions: 1, standard proteins; 2, kamaboko gel of WM; 3, setting gel of WM; 4, modori gel of WM; 5, kamaboko gel of AC; 6, setting gel of AC; 7, modori gel of AC; 8, kamaboko gel of AK; 9, setting gel of AK; and 10, modori gel of AK. WM: water-washed surimi, AC: acid-isolated protein, AK: alkali-isolated protein.

**Table 1 foods-12-00939-t001:** The effect of the process on the protein recovery rate and residue activity of cathepsin B and cathepsin L.

	Protein Recovery (%)	Cathepsin B (U/g)	Cathepsin L (U/g)	GEO (ug/kg)	MIB (ug/kg)
Muscle	^_^	152.33 ± 8.21 ^a^	92.51 ± 6.20 ^a^	1.51 ± 0.08 ^a^	2.06 ± 0.11 ^a^
WM	28.8 ± 0.87 ^b^	7.79 ± 2.13 ^c^	29.14 ± 1.59 ^c^	0.92 ± 0.05 ^b^	1.02 ± 0.05 ^b^
AC	38.5 ± 0.59 ^a^	25.58 ± 4.85 ^b^	65.43 ± 4.91 ^b^	0.35 ± 0.03 ^c^	0.33 ± 0.03 ^c^
AK	40.9 ± 1.24 ^a^	8.53 ± 2.60 ^c^	27.67 ± 2.38 ^c^	0.26 ± 0.04 ^d^	0.21 ± 0.03 ^d^

^a–d^ Means in each column with different superscript letters are significantly different (*p* < 0.05) (n = 3). GEO indicates geosmin, MIB indicates 2-methylisoborneol. WM: water-washed surimi, AC: acid-isolated protein, AK: alkali-isolated protein.

**Table 2 foods-12-00939-t002:** TCA-soluble peptide content (mg/g) and disulfide content (mol/10^7^ g protein) of WM, IPs and gels prepared under different conditions.

Samples	Peptide Content (mg/g)			S-S Content (mol/10^7^ g Protein)		
	WM	AC	AK	WM	AC	AK
Surimi/IPs	31.71 ± 1.32 ^c^	46.85 ± 2.50 ^a^	36.37 ± 1.87 ^b^	2.49 ± 0.12 ^C^	3.97 ± 0.18 ^B^	4.46 ± 0.21 ^A^
Kamaboko	44.18 ± 2.41 ^b^	61.42 ± 3.08 ^a^	47.05 ± 1.35 ^b^	15.26 ± 1.10 ^A^	15.01 ± 0.81 ^A^	15.14 ± 1.25 ^A^
Setting gel	52.56 ± 2.64 ^c^	77.55 ± 2.52 ^a^	61.27 ± 3.99 ^b^	15.35 ± 0.96 ^A^	15.30 ± 0.64 ^A^	15.42 ± 0.77 ^A^
Modori gel	62.92 ± 2.86 ^c^	90.89 ± 4.65 ^a^	76.84 ± 2.15 ^b^	15.18 ± 1.05 ^A^	14.81 ± 1.32 ^A^	15.20 ± 1.19 ^A^

^a–c^ Means in each row of peptide content with different superscript letters are significantly different (*p* < 0.05) (n = 3). ^A–C^ Means in each row of S-S content with different superscript letters are significantly different (*p* < 0.05) (n = 3). WM: water-washed surimi, AC: acid-isolated protein, AK: alkali-isolated protein, IPs: the isolated proteins.

**Table 3 foods-12-00939-t003:** Breaking force and breaking deformation of gel prepared under different processes and conditions.

Surimi/IPs		Breaking Force (g)	Deformation (mm)
WM	Kamaboko gel	467.2 ± 20.6 ^b^	12.1 ± 0.3 ^b^
	Setting gel	489.1 ± 16.1 ^b^	12.3 ± 0.3 ^b^
	Modori gel	315.9 ± 18.8 ^e^	10.8 ± 0.5 ^de^
AC	Kamaboko gel	347.5 ± 24.7 ^e^	11.7 ± 0.3 ^cd^
	Setting gel	302.6 ± 20.4 ^e^	10.3 ± 0.4 ^e^
	Modori gel	226.2 ± 19.5 ^f^	8.3 ± 0.4 ^f^
AK	Kamaboko gel	435.8± 24.6 ^c^	11.9 ± 0.4 ^bc^
	Setting gel	518.9 ± 21.4 ^a^	12.8 ± 0.3 ^a^
	Modori gel	386.4 ± 15.7 ^d^	11.6 ± 0.2 ^cd^

^a–f^ Means in each column with different superscript letters are significantly different (*p* < 0.05) (n = 5). WM: water-washed surimi, AC: acid-isolated protein, AK: alkali-isolated protein, IPs: the isolated proteins.

## Data Availability

Anybody need the data could contact the corresponding author.

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
