# Peer review of "Effect of pH-Shifting Process on the Cathepsin Activity, Muddy Off-Odor Compounds’ Content and Gelling Properties of Isolated Protein from Silver Carp"

_foods, 2023, doi:10.3390/foods12050939_

Round 1
Reviewer 1 Report
The paper represents the effect of pH-shifting process on the cathepsin activity, content of muddy off-odor compounds and gelling properties of isolated protein from silver carp. However, the authors did not explain the reason why the pH-shifting affected these indicators. Also, they did not clarify the significance of their work. Honestly, I think that some data is missing in order to support the adequacy of the proposed method so that I cannot recommend the publication of the manuscript in its actual state.
Specific Comments:
1. The method of pH shifting process in the manuscript is highlighted, but the author has not explained clearly about the shift process. Please give detailed explanation?
2. The abstract is not well organized and logically confusing. It needs to be rewritten.
3. The conclusion of the manuscript clarifies the advantages of acid-isolated protein. What is the significance of pH shifting process?
4. Why the pH shifting process used to isolate protein from surimi will affect the content of these two odor substances?
5. Is the muddy off-odor only caused by GEO and MIB? Please check carefully.
6. Line 142:The content determination process of two substances (GEO and MIB) is not described clearly in the manuscript. How to determine the content, using standards or other methods?
7. Line 145:Please add specific experimental conditions, including type of capillary column, column temperature and instrument condition.
8. All used abbreviations must be defined when first used. There are abbreviations that are very familiar to science readers, but it must be considered that the work may have readers from different areas. GEO, MIB and many other abbreviations must be defined.
9. Line 14, 24, 127,243 and 248: The font size of the manuscript is not uniform. Please check the whole manuscript.
10. Line 105 : It should be 4℃, NOT 4 oC. Please check the whole manuscript.
11. Line 166 : It should be clear about the size of the sample used, the model of the texture analyzer probe and the operating depth.
12. Line 181 : Clarify the sample concentration and loading volume.
13. Line207-218: The percentage reduction of GEO and MIB in the acid- and base-extracted proteins in this passage does not match the data shown in Table 1.
14. Table 1 lacks explanatory notes for each abbreviated name in the table.
15. Line259: The writing format of p < 0.05 in the manuscript is not uniform, please check.
16. Line263: The color of table lines in Figure 2 is not uniform.
17. Line281: Why does the author determine that the labeled protein band is MHC without the support of experimental data.
18. The way to include the bibliographical references in the manuscript must comply with the journal's standards [1], [2,3], [4-6]. Please correct throughout the manuscript.
19. The bibliographical references section must be corrected according to the rules for authors. For example, the journal name should be abbreviated. « Ávila-Román, J.; Soliz-Rueda, J.R.; Bravo, F.I.; Aragones, G.; Suarez, M.; Arola-Arnal, A.; Mulero, M.; Salvadó, M.J.; Arola, L.; Torres-Fuentes, C.; et al. Phenolic Compounds and Biological Rhythms: Who Takes the Lead? Trends Food Sci. Technol. 2021, 113, 77–85”.

Author Response
请参阅附件

Reviewer 2 Report
Comments on “Effect of pH-shifting process on the cathepsin activity, muddy off-odor compounds’ content and gelling properties of isolated protein from silver carp”
1. Line 12. Please italicize the scientific name.
2. Line 15. It should read as “the conventional cold water washing”
3. How about the removal efficacy of lipid and heme proteins? Those components and their products can intensify the off-odor of the fish flesh.
4. Line 26. Please state the setting condition.
5. Check the reference style throughout the manuscript.
6. Line 47-50. Please add the scientific name of all species.
7. Line 81-83. Please add the reference “Phetsang, H., Panpipat, W., Panya, A., Phonsatta, N., & Chaijan, M. (2021). Occurrence and development of off-odor compounds in farmed hybrid catfish (Clarias macrocephalus× Clarias gariepinus) muscle during refrigerated storage: Chemical and volatilomic analysis. Foods, 10(8), 1841.”
8. Line 92-93. Because the authors employed live fish, please provide animal ethics.
9. Line 112. How long did it take for ice storage?
10. Line 127. Check the spelling “quality”
11. Line 158. Why sorbitol and sucrose were added at this step? They are cryoprotactant and should be used for the frozen surimi/PIs.
12. Line 181-186. Please state “reducing” or “non-reducing condition” for SDS-PAGE and also please provide the staining and de-staining procedures.
13. Line 198. Use "pH-shift" rather than "new" because it is not a new method; otherwise, you will have modified the pH-shift process.
14. Line 215-216. What about the lipid content in this research? Additionally, both geosmin and MIB can degrade during pH-shift processing. This will also result in a decrease in detected contents.
15. Line 282. Delete “And” and start the sentence with “The decrease…”
16. Table 1 and Table 3. Please provide the full name of all abbreviations in the footnote.
17. Line 336. Did you test for endogenous TGase activity? If not, please revise the statement to avoid overstating it. Please also check in the Discussion.
18. Sensory evaluation (i.e. muddy odor/off-odor) of the gel is missing.
19. Line 337-339. How about the degree of denaturation of proteins under pH-shift conditions? It can also affect the gel-forming ability of PIs.
20. Water holding capacity and whiteness of gels are needed.
21. Fig 1. How about the G” and Tand?
22. Unwashed mince should be used to compare for all parameters.
Author Response
请参阅附件

Round 2
Reviewer 1 Report
The paper represents the effect of pH-shifting process on the cathepsin activity, content of muddy off-odor compounds and gelling properties of isolated protein from silver carp. This has a positive contribution to the study of silver carp processing and its flavor quality control. Please check abstract carefully, and improve English expression.
Author Response
Dear Reviewer,
We are truly grateful to you for giving us the chance to revise our manuscript. Based on the reviewer’s comments and suggestions, we have made careful modifications on the abstract part of original manuscript and the language of the full paper. All changes made to the text are in red color.
Reviewer 2 Report
All points raised by reviewers were addressed and answered point-by-point. So, it can be accepted.
Author Response

(The authors gave the same response as above.)
